

# Assessing the Performance of Troposphere Tomographic Modeling Using Multi-Source Water Vapor Data During Hong Kong's Rainy Season from May to October 2013

Biyan Chen, Zhizhao Liu

Department of Land Surveying & Geo-Informatics, Hong Kong Polytechnic University, Hong Kong, China

*Correspondence to*: Zhizhao Liu (lszzliu@polyu.edu.hk)

**Abstract**: Acquiring accurate atmospheric water vapor spatial information remains one of the most challenging tasks in meteorology. Tomographic technique is a powerful tool to model atmospheric water vapor and monitor the water vapor spatial and temporal distribution/variation information. This paper presents a study on monitoring of water vapor variations

using tomographic technique based on multi-source water vapor data, including GPS (Global Positioning System), radiosonde, WVR (Water Vapor Radiometer), NWP (Numerical Weather Prediction), AERONET (AErosol RObotic NETwork) sun photometer and synoptic stations. An extensive investigation has been carried out using multi-source data collected from May to October 2013 in Hong Kong. With the use of radiosonde observed profiles, five different vertical a priori information schemes were designed and examined. Analysis results revealed that the best vertical constraint is to

employ the average radiosonde profiles over the three days prior to the tomographic time and that the assimilation of multi-source data can increase the tomography modeling accuracy. Based on the best vertical a priori information scheme, comparisons of SWD measurements between GPS and multi-observational tomography showed that the RMS error of their differences is 10.85 mm. Multi-observational tomography achieved an accuracy of 7.13 mm/km when compared with radiosonde wet refractivity observations. The vertical layer tomographic modeling accuracy was also assessed using

radiosonde water vapor profiles. An accuracy of 11.44 mm/km at the lowest layer (0 to 0.4 km) and an RMS error of 3.30 mm/km at the uppermost layer (7.5 km to 8.5 km) were yielded. At last, a test of the tomographic modeling in a torrential storm occurring on 21~22 May 2013 in Hong Kong demonstrated that the tomographic modeling is very robust even during severe precipitation conditions.

**Keywords**: Water vapor, tomography, wet refractivity, multi-source data

# 1 Introduction

Water vapor is a potent greenhouse gas in the earth's atmosphere and plays an important role in many atmospheric processes. It contributes significantly to the formation of many weather phenomena such as cloud, rain, snow, sleet, hail and other precipitations. A small amount of water vapor variation may cause severe weather changes (Mohanakumar, 2008). Accurate





information of water vapor spatiotemporal distributions is thus crucially important for weather forecasting services and meteorological research, such as precipitation and severe weather forecast, and natural hazard mitigation (Bender and Raabe, 2007; Perler et al., 2011; Rocken et al., 1997). However, atmospheric water vapor remains one of the most poorly characterized parameters in meteorology due to its high variability nature in space and time (Lee et al., 2013; Rocken et al., 1997).

Over the past years, many techniques have been developed to improve the observation of atmospheric water vapor, including both ground-based observation systems and satellite-borne remote sensing sensors (Guiraud et al. 1979; Elgered et al. 1991; Holben et al. 2001; Niell et al. 2001; Gao and Kaufman 2003). Among various platforms, Global Navigation Satellite System (GNSS) has been considered as a powerful approach to retrieve atmospheric water vapor data with high spatial and

temporal resolutions. In addition, GNSS also has the advantages of low operational cost and all-weather capability when compared to other traditional means. For example, limited by the high expense of launching weather balloons, there are only about 850 radiosonde sites globally and radiosonde measurements are usually made only twice per day at most stations (Kuo et al., 2005; Niell et al., 2001). The poor regional coverage and low temporal resolution of the radiosonde observations significantly limit their values in many applications such as weather forecasting. Another important instrument for water

vapor measurement is water vapor radiometer (WVR) that has been often used to correct tropospheric wet delay in geodetic observations such as very-long-baseline interferometry (VLBI) (Beckman, 1985; Elgered et al., 1991). However, WVR is sensitive to weather conditions and large uncertainties may exist when observation is made in rainy or foggy conditions. The strengths of GNSS in atmospheric sounding have significantly facilitated the development of GNSS meteorology, which has become a focus of multi-disciplinary research in the fields of meteorology and space geodesy.

The concept of GNSS meteorology was first documented in Bevis et al. (1992) in which the possibilities of Global Positioning System (GPS) remote sensing of atmospheric water vapor were elaborated. Thereafter numerous field campaigns demonstrated the GPS/GNSS ability to accurately measure atmospheric water vapor and the derived precipitable water vapor (PWV) data can reach an accuracy of 1~2 mm or even better (Duan et al., 1996; Elgered et al., 1997; Lee et al., 2013; Liu and Li, 2013; Rocken et al., 1993; Tregoning et al., 1998). GNSS-inferred PWV data have enriched the meteorological

research by providing detailed information of horizontal distribution of atmospheric water vapor. However, the vertical profile information remains unknown. Inspired by the capability of tomography technique of reconstructing three-dimensional field, Bevis et al. (1992) also envisioned the potential of tomographic technique in the reconstruction of 3D water vapor distribution using GPS-derived slant wet delay (SWD) data. In 2000, Flores et al. (2000) performed an experiment of water vapor tomography based on a GPS network in Hawaii, USA. This was the first time that tomographic

technique was demonstrated to reconstruct 3D structure of tropospheric water vapor. After this successful experiment, more work in tropospheric tomography has been carried out in the GPS/geodesy community (Champollion et al., 2005; Bender and Raabe, 2007; Rohm and Bosy, 2009; Notarpietro et al., 2011; Perler et al., 2011; Bender et al., 2011; Rohm and Bosy, 2011). Bi et al., (2006) carried out a water vapor tomography experiment by using a small GPS network in the Beijing region. The accuracy of wet refractivity profiles from tomographic solution can reach ~7 mm/km by comparing with radiosonde



ones. Troller et al., (2006) investigated the tomographic technique using GPS observations from the Swiss national GPS network AGNES of the Swiss Federal Office of Topography. Comparisons of water vapor profiles between tomography and numerical weather models showed that the RMS error can reach an order of better than 10 mm/km. Xia et al., (2013) presented a study for water vapor tomography using GPS observations and radio occultation profiles. An overall accuracy of 6.3 mm/km of tomographic results is achieved for a 10-day test. In the research reported by Shangguan et al., (2013), GPS tomography results in a whole year 2007 were evaluated using radiosonde data and wet refractivity field of accuracy 6.5~9.0 mm/km is obtained. A one-year tomography experiment in Hong Kong was carried out by Jiang et al., (2014), tomographic result of accuracy ~7.9 mm/km has been obtained when compared with radiosonde data.

However, some limitations in the tomographic technique still have not been resolved (Bender et al., 2009; Bender and Raabe, 2007; Rohm et al., 2013). In the tomographic approach, the probed space is usually discretized into a number of 3D closed voxels. Water vapor quantity in each voxel can then be estimated from a large number of integral water vapor ray paths using the tomographic technique. This requires each voxel be crossed by a number of GNSS signals from different directions. In practice, this requirement is hardly satisfied because: (1) most GNSS networks are not dedicatedly designed for tomography purpose. To ensure each voxel being crossed by GNSS signals requires a high density of GNSS receivers in the network, which is practically impossible for cost and operational reasons; (2) at present, the number of trackable GNSS satellites during a tomographic period is limited, which restricts the number of rays that cross through the voxels; this situation is expected to improve with the launch of more satellites in Beidou and Galileo navigation satellite systems; and (3) water vapor is highly variable in both spatial and temporal domains, thus the voxel size should not be too large spatially and the tomographic period should not be too long temporally. As a result, it is almost impossible to tomographically reconstruct a 3D water vapor field by using GNSS data alone. This problem can be resolved by adding inter-voxel constraints and especially by introducing non-GNSS measurements (Bender and Raabe, 2007; Bevis et al., 1992). Several studies have shown that GNSS tropospheric tomography has improved after assimilating other observations, such as radiosonde (Bi et al., 2006; Skone and Hoyle, 2005), numerical weather prediction (NWP) (Notarpietro et al., 2011) and radio occultation (Xia et al., 2013).

In the past studies, the type of water vapor data sources used in tomography is still very limited, usually from one single type of water vapor observation technique. In this study, we will investigate the tomographic technique by assimilating water vapor measurements from six sources available in Hong Kong region. In addition to GPS, water vapor from other five sources are also used, namely radiosonde, WVR, NWP, AErosol RObotic NETwork (AERONET) sun photometer and synoptic stations. Radiosonde water vapor data provide excellent vertical profile observation information, which is crucial for tomographic modeling. The availability of abundant non-GNSS data that are of different characteristics offers us the opportunity to examine their contribution to water vapor tomography results. In this study, we will investigate approaches of how to properly assimilate these data into the tomographic model. Five schemes that contain different vertical a priori information are designed and examined. The performance of the multi-observational tomography is fully evaluated using GNSS data and radiosonde profiles. In addition, the tomographic results are applied to reveal the evolution of wet





refractivity field during heavy precipitation event. This paper is structured as follows. Section 2 provides an overview of multiple water vapor observation systems in Hong Kong. A description of the principle of water vapor tomography with multi-source data is presented in Section 3. Section 4 is dedicated to the evaluation of the performance of water vapor tomography. Conclusions and final remarks are given in Section 5.

## 5  2 Description of tomography inputs

In this study, water vapor data for tomographic modeling are obtained from GPS, radiosonde, WVR, NWP, AERONET and synoptic stations. Figure 1 shows the geographical distribution of GPS, radiosonde, WVR, AERONET stations in Hong Kong. Actually, the synoptic stations are collocated with the GPS stations (a total of 12 stations). Each GPS station is equipped with meteorological instruments to record air pressure, temperature and relative humidity. Refractivity computed

10  from these parameters (more details in Sect. 3) can be used as good input data in the tomography.

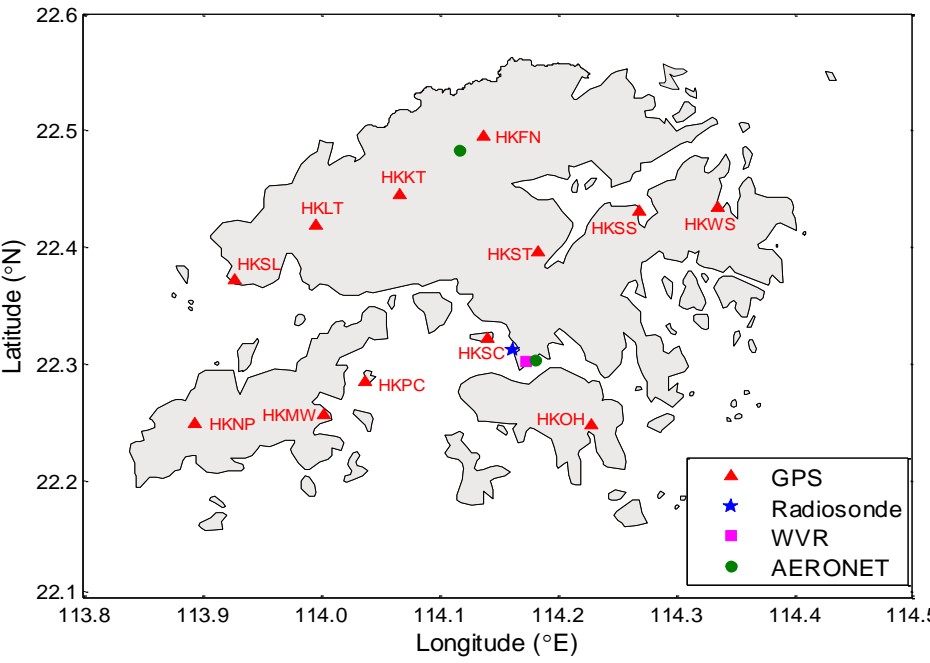

**Figure 1: Geographical distribution of GPS, radiosonde, WVR and AERONET stations in Hong Kong**

### 2.1 GPS observations

15  The Lands Department of the Government of Hong Kong Special Administrative Region (HKSAR) has been operating the GPS network – Hong Kong Satellite Positioning Reference Station Network (SatRef) since 2000 (Chan and Li, 2007). Before 2015, this network consists of 12 GPS stations and their locations are shown in Figure 1. GPS signals are significantly affected when they traverse the neutral atmosphere. The tropospheric path delay is a major error source in GPS



precise positioning. Usually, the tropospheric delay can be divided into hydrostatic and wet components and the wet component can be estimated together with GPS coordinate parameters. Currently, many GNSS data processing software packages are capable of accurately estimating the tropospheric delay. In this study, we adopt the Bernese GNSS software to process the GPS data. This software uses double-difference to remove the satellite and receiver clock biases and outputs

many products including zenith tropospheric delay (ZTD), gradients and the double-differenced residuals (Dach et al., 2007). The slant wet delays can thus be retrieved according to (Chen and Liu, 2014)

$$\text{SWD} = (\text{ZTD} - \text{ZHD}) \cdot f(z) + \frac{\partial f}{\partial z} \cdot \left( G_{N,W} \cdot \cos(\phi) + G_{E,W} \cdot \sin(\phi) \right) + R \qquad (1)$$

where ZHD is the zenith hydrostatic delay which can be accurately modeled with surface meteorological observations. $z$ and $\phi$ are satellite zenith distance and azimuth angle, respectively. $f$ is the wet mapping function. In our GNSS data processing,

the wet Niell mapping function (Niell, 1996) is used. $G_{N,W}$ and $G_{E,W}$ are the wet delay gradient components in the northern and eastern directions. The last term $R$ refers to the post-fit residuals.

## 2.2 WVR observations

One water vapor radiometer (WVR), which is located at the Hong Kong Observatory (HKO) (shown in Figure 1), is used for this study. This WVR uses seven oxygen channels and five water vapor channels to make observations of the temperature,

humidity and liquid water vapor profiles up to 10 km above the ground in the zenith mode (Chan, 2010). The HKO employs neural network approach and radiosonde profiles to establish a statistical model between the WVR brightness temperature and the vertical profiles of temperature and relative humidity (Chan, 2010). Based on this statistical model, temperature and relative humidity profiles can be retrieved from the WVR's brightness temperature measurements. The WVR data used in this study have a temporal resolution of 15 minutes.

## 2.3 Water vapor data derived from the NWP

NWP non-hydrostatic model provides a good means to investigating small scale meteorological phenomena (Saito, 2007). On 1 September 2004, the Japan Meteorological Agency (JMA) started to run a non-hydrostatic model with a horizontal resolution of 10 km to support weather disaster prevention (Saito et al., 2006). Based on the successful trials of using the JMA non-hydrostatic model, HKO has been operating a new NWP system since 2010 (Chan et al., 2010). This system has

the ability to perform predictions at a horizontal resolution of 2 km and a temporal solution of one hour (Wong, 2010). The domain of this model is 608 km × 608 km which covers Hong Kong and its surrounding regions. It can output several parameters such as temperature, dew point depression and geopotential height at 16 isobaric levels ranging from 1,000 hPa to 100 hPa at the top level. However, NWP data have a limited precision because it is predicted based on physical principles rather than real observations.



In tomographic modeling, a considerable amount of SWD that do not fully traverse the tomographic volume are not used (see the dashed rays that cross the grey shaded area in Figure 2). However, these SWD data (especially at low elevations) are helpful to improve the lower layers reconstruction (Notarpietro et al., 2011). In order to make a full use of the SWDs, the SWDs that partially pass through the tomographic modeling area are divided into two parts. As shown in Figure 2, the SWD inside the tomography volume is called $SWD_{in}$ and the rest that is outside the modeling area is referred to as $SWD_{out}$. The $SWD_{out}$ cannot be used for the tomographic modeling since it is outside the modeling region. In this study, the $SWD_{out}$ is calculated from the NWP profile data. After subtracting the $SWD_{out}$ from the SWD, the $SWD_{in}$ can be derived and will be used in the tomographic modeling process.

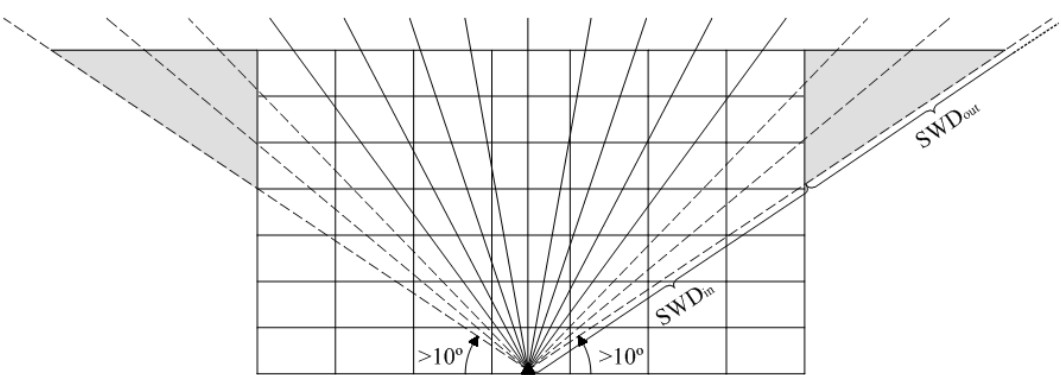

**Figure 2: Schematic diagram of rays used in the tomography. The rectangle defines the tomographic region. Normally, only rays (in solid lines) that enter the tomographic model from the top layer can be used and rays (in dashed lines) enter from the laterals should be rejected.**

## 2.4 AERONET observations

AERONET is a ground-based network consisting of more than 300 globally distributed sun photometers that are mainly used to study atmospheric aerosol properties (Holben et al., 1998; Liu et al., 2013b). The sun photometers are able to make direct solar extinction measurements at multiple wavelengths ranging from 340 to 1640 nm with an interval of 15 min (Giles et al., 2012; Holben et al., 2001). The observations made at the wavelength of 940 nm can be employed to retrieve water vapor (Holben et al., 2001). At present, there are two AERONET sun photometers operating in Hong Kong (as seen in Figure 1). Liu et al., (2013a) did an assessment of 6 years of water vapor measurements recorded by the AERONET station in Hong Kong. Their study demonstrated that AERONET sun photometer can provide accurate precipitable water vapor measurements and that the agreement with radiosonde water vapor data was 2.89 mm in RMS error. Thus, AERONET can be a good data source for water vapor tomography. One drawback of sun photometer is that it can work only in the periods with direct sunlight. No data are available at the nighttime or in condition of precipitation.





## 2.5 Vertical a priori information from radiosonde profiles

With sensors ascending together with weather balloon, radiosonde can make meteorological observations including pressure, temperature and relative humidity at various heights (World Meteorological Organization, 2008). This enables us to get accurate wet refractivity profiles from radiosonde observations. In Hong Kong, there is one radiosonde station located at the

King's Park (22.31 °N, 114.17 °E) and this station is operated by the HKO. Radiosonde balloon is launched twice daily at UTC 00:00 and 12:00, respectively. Water vapor profiles retrieved from radiosonde are often adopted as vertical a priori information in water vapor tomography (Bi et al., 2006; Champollion et al., 2009; Skone and Hoyle, 2005). Good a priori water vapor information can significantly improve tomographic results especially for flat regions (Notarpietro et al., 2011). Hong Kong is a relatively flat region. The largest altitude difference among the 12 GNSS stations is only about 330 m. It is

therefore very crucial to impose good a priori vertical information for water vapor tomographic modeling in the Hong Kong region. HKO has archived a long time series of water vapor profile records. By statistical analysis of the Hong Kong radiosonde profiles over the 10 years (2003-2012), a priori information of wet refractivity vertical distribution in Hong Kong is derived. In this tomographic study, we are going to evaluate the impact of five schemes of different a priori vertical information on the tomographic modeling solutions. The details of the five schemes are described as follows:

V1. In our tomography model, the troposphere is divided into 15 non-uniform layers (more details in Sect. 3). Water vapor profile for each vertical layer is averaged from three-day radiosonde observations prior to the tomographic modeling;

V2. For each vertical layer, the a priori wet refractivity value is averaged from 10 years (2003-2012) of radiosonde data. Meanwhile, a statistical variance-covariance matrix for the a priori information is generated from the 10-year radiosonde wet refractivity profiles, which will be used to determine the weight matrix for the vertical a priori information in the

tomography;

V3. Similar to V2, statistics are performed with the 10 years of radiosonde data for every month. Therefore, one mean value and one statistical variance-covariance matrix can be derived for each month. In the tomography, a priori information corresponding to the tomographic modeling month is employed;

V4. Different from V1 to V3, ratio of wet refractivity between each two neighboring layers are used as a priori information.

For each pair of neighboring vertical layers, average ratio of their wet refractivities is derived from the 10 years of radiosonde profiles. Also a statistical variance-covariance matrix for the ratios can be calculated;

V5. Similar to V4, statistics are performed with the 10 years of radiosonde data for every month. Therefore, for each month, a pair of average ratio value and statistical variance-covariance matrix is derived. Same as V3, a priori information corresponding to the tomographic modeling month is employed in the tomography.

## 3 Water vapor tomography with multi-source data


When GNSS radio signals propagate through the troposphere, the signals are delayed due to the refraction of water vapor. The excess path experienced by the radio signals is often referred to as tropospheric wet delay, which can be expressed as:





$$SWD = 10^{-6} \int_l N_w d_l \qquad (2)$$

where $N_w$ represents the wet refractivity and $l$ is the ray path of the radio signal through the troposphere. The wet refractivity is a function of the partial pressure of water vapor $e$ (unit: hPa) and the temperature $T$ (unit: Kelvin degree) (Rüeger, 2002; Smith and Weintraub, 1953)

$$5 \quad N_w = 22.9721\frac{e}{T} + 375463\frac{e}{T^2} \qquad (3)$$

The wet refractivity is an important parameter describing the water vapor distribution in the atmosphere. According to Eq. (3), the wet refractivity of a certain point can be obtained by measuring the ambient air pressure and temperature. However, it is difficult to acquire meteorological observations in the upper atmosphere. Developing a tomographic modeling approach to characterize water vapor 3D spatial distribution is therefore highly desired.

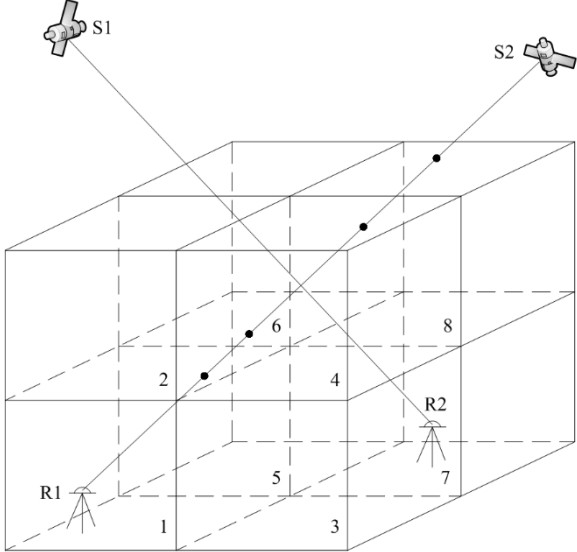

**Figure 3: Schematic representation of three-dimensional water vapor tomography**

In Figure 3, the probed tropospheric region is divided into 8 voxels with the assumption that the wet refractivity inside each

15    voxel is invariable during the tomographic modeling period. Examining the R1-S2 ray path, it can be observed that it passes through 4 voxels numbered as 1, 2, 4 and 8. In the tomographic technique, the SWD should be equal to the summation of the product of wet refractivity and the length of ray path within each voxel. For the R1-S2 ray, we can thus get:

$$SWD_{R1-S2} = a_1 \cdot x_1 + a_2 \cdot x_2 + a_4 \cdot x_4 + a_8 \cdot x_8 \qquad (4)$$





where $a_i$ ($i=1,2,4,8$) represents the length of ray intercepted by voxel $i$, $x_i$ stands for the wet refractivity in voxel $i$. Actually, Eq. (4) is the linear form of Eq. (2). During a tomographic process, a lot of ray paths linking GNSS satellites and ground GNSS receivers will traverse the 3D modeling voxels. Thus, the Eq. (4) can be rewritten in a matrix form:

$$\mathbf{y} = \mathbf{A} \cdot \mathbf{x} \tag{5}$$

where $\mathbf{y}$ is the vector of water vapor observations, e.g., the slant wet delays derived from GNSS; $\mathbf{x}$ is the vector of unknown wet refractivity of each voxel; $\mathbf{A}$ represents the matrix describing the path length of each signal intercepted by each voxel. It should be noted that the wet refractivity field can hardly be inverted by Eq. (5), as not all voxels are crossed by GNSS satellite signals. To overcome the rank-defect problem, extra water vapor observations and constraints are needed. As described in Sect. 2, WVR, NWP, AERONET and synoptic stations can also provide water vapor measurements. In addition,

the vertical a priori information derived from radiosonde profiles and horizontal smoothing constraint are augmented to Eq. (5) to increase the rank of matrix $\mathbf{A}$. The horizontal constraint is added based on the assumption that wet refractivity in a voxel is the weighted average of its horizontal neighbors (Flores et al., 2000). Combining all available observations and constraints, the tomography Eq. (5) becomes:

$$
\begin{pmatrix}
\mathbf{y}_\mathrm{G} \\
\mathbf{y}_\mathrm{W} \\
\mathbf{y}_\mathrm{N} \\
\mathbf{y}_\mathrm{A} \\
\mathbf{y}_\mathrm{s} \\
\mathbf{y}_\mathrm{R} \\
\mathbf{0}
\end{pmatrix}
=
\begin{pmatrix}
\mathbf{A}_\mathrm{G} \\
\mathbf{A}_\mathrm{W} \\
\mathbf{A}_\mathrm{N} \\
\mathbf{A}_\mathrm{A} \\
\mathbf{A}_\mathrm{s} \\
\mathbf{A}_\mathrm{R} \\
\mathbf{H}
\end{pmatrix}
\cdot \mathbf{x}
\tag{6}
$$

where $\mathbf{y}_\mathrm{G}$, $\mathbf{y}_\mathrm{W}$, $\mathbf{y}_\mathrm{N}$, $\mathbf{y}_\mathrm{A}$, $\mathbf{y}_\mathrm{s}$ and $\mathbf{y}_\mathrm{R}$ refer to the water vapor data derived from GPS, WVR, NWP, AERONET, synoptic observations and radiosonde, respectively; $\mathbf{A}$ with subscripts represents coefficient matrix for each type of data; $\mathbf{H}$ is the coefficient matrix for the horizontal constraint. By performing the Least Squares method, the wet refractivity of all the voxels can be solved:

$$\mathbf{x} = \left[ w_1 \cdot \left( \mathbf{A}_\mathrm{G}^\mathrm{T} \cdot \mathbf{P}_\mathrm{G} \cdot \mathbf{A}_\mathrm{G} + \mathbf{A}_\mathrm{W}^\mathrm{T} \cdot \mathbf{P}_\mathrm{W} \cdot \mathbf{A}_\mathrm{W} + \mathbf{A}_\mathrm{A}^\mathrm{T} \cdot \mathbf{P}_\mathrm{A} \cdot \mathbf{A}_\mathrm{A} + \mathbf{A}_\mathrm{s}^\mathrm{T} \cdot \mathbf{P}_\mathrm{s} \cdot \mathbf{A}_\mathrm{s} + \mathbf{A}_\mathrm{R}^\mathrm{T} \cdot \mathbf{P}_\mathrm{R} \cdot \mathbf{A}_\mathrm{R} \right) + w_2 \cdot \mathbf{A}_\mathrm{N}^\mathrm{T} \cdot \mathbf{P}_\mathrm{N} \cdot \mathbf{A}_\mathrm{N} + w_3 \cdot \mathbf{H}^\mathrm{T} \cdot \mathbf{P}_\mathrm{H} \cdot$$

$$\mathbf{H} \right]^{-1} \cdot \left[ w_1 \cdot \left( \mathbf{A}_\mathrm{G}^\mathrm{T} \cdot \mathbf{P}_\mathrm{G} \cdot \mathbf{y}_\mathrm{G} + \mathbf{A}_\mathrm{W}^\mathrm{T} \cdot \mathbf{P}_\mathrm{W} \cdot \mathbf{y}_\mathrm{W} + \mathbf{A}_\mathrm{A}^\mathrm{T} \cdot \mathbf{P}_\mathrm{A} \cdot \mathbf{y}_\mathrm{A} + \mathbf{A}_\mathrm{s}^\mathrm{T} \cdot \mathbf{P}_\mathrm{s} \cdot \mathbf{y}_\mathrm{s} + \mathbf{A}_\mathrm{R}^\mathrm{T} \cdot \mathbf{P}_\mathrm{R} \cdot \mathbf{y}_\mathrm{R} \right) + w_2 \cdot \mathbf{A}_\mathrm{N}^\mathrm{T} \cdot \mathbf{P}_\mathrm{N} \cdot \mathbf{y}_\mathrm{N} \right] \tag{7}$$

where $w_1$, $w_2$ and $w_3$ are weighting factors that will be discussed later; $\mathbf{P}$ with subscripts represents weight matrix for each type of data and constraints. The both $\mathbf{P}_\mathrm{G}$ and $\mathbf{P}_\mathrm{N}$ weight matrices are diagonal with elements defined as $\sin(\theta)$ ($\theta$ refers to the elevation angle of the SWD). This is based on the fact that the error in SWD usually increases when the elevation angle decreases. For the weight matrices $\mathbf{P}_\mathrm{W}$, $\mathbf{P}_\mathrm{A}$, $\mathbf{P}_\mathrm{s}$ and $\mathbf{P}_\mathrm{H}$, they are defined as unit matrices. As mentioned in Sect. 2.5, the

weight matrix $\mathbf{P}_\mathrm{R}$ is from the statistical variance-covariance matrix that derived from the radiosonde profiles. The three weighting factors $w_1$, $w_2$ and $w_3$ in Eq. (7) are determined by using the Helmert variance component estimation method (Kizilsu and Sahin, 2000; Wang et al., 2009). The reasons for categorizing the GPS, WVR, AERONET, synoptic





observations and radiosonde into one group are: 1) water vapor measurements from these techniques are at a similar level; 2) the number of observations from WVR, AERONET and synoptic stations is much smaller compared with GPS data. Since the NWP data have lower accuracy than GPS ones and tight horizontal constraint can result in a very smooth water vapor distribution in the horizontal direction, two weighting factors are assigned to adjust their impact on the result. Actually, our tomographic experiments show that $w_2$ and $w_3$ are always smaller than $w_1$, which implies that the impact of NWP data and horizontal constraint to the solution is degraded. Since the wet refractivity field obtained from Eq. (7) is just an approximate solution, the multiplicative algebraic reconstruction technique (MART) is finally implemented to improve the wet refractivity solution from Eq. (7) (Bender et al., 2011; Chen and Liu, 2014). The Least Squares solution of Eq. (7) provides an initial state to the MART algorithm to converge, which will produce a more accurate wet refractivity filed. The advantages of this combined reconstruction algorithm have been demonstrated in several studies (Notarpietro et al., 2011; Wen et al., 2008; Xia et al., 2013). In this study, tomographic model is discretized using method developed in (Chen and Liu, 2014). In the horizontal, resolutions of 0.08° (about 8.5 km) are set for both latitude and longitude directions. The top boundary of 8.5 km is adopted for the tomography model (Liu et al., 2014). From the surface to the top, the troposphere is divided into 15 non-uniform layers in the vertical direction (Chen and Liu, 2014). From ground upward, the layer thickness is arranged as: 400 m for the bottom 5 layers, 500 m for the 4 layers, 600 m for the 3 layers, 700 m for the 1 layer, and 1,000 m for the top 2 layers.

## 4 Analysis of the tomographic results

Many tests have been carried out to evaluate the performance of the above water vapor tomographic model. The multi-source data used in the tests were collected from May to October 2013, the most humid period in a year in Hong Kong. Severe weathers such as typhoons and rainstorms often occur in these months. Assessing the model's performance of retrieving spatial distribution and temporal variation of atmospheric water vapor under severe weather conditions is particularly interesting to us because 3D water vapor distribution and propagation information can provide valuable assistance to weather forecasters. In this study, tomography is performed consecutively with an interval of 30 min. In order to identify the best vertical a priori information, the five different schemes as described in Sect. 2.5 are used. SWD data from the GPS observations of HKLT station are used for quality assessment and thus are not used in the tomographic modeling. In addition, radiosonde profiles are also exploited to assess the tomographic vertical distribution of wet refractivity.

### 4.1 Water vapor tomographic performance using multi-source data

Once we obtain the tomographic wet refractivity field from Eq. (7), SWD along a specific ray path can be derived by an integral of the GPS path length and water vapor refractivity in each voxel. These tomographic SWDs can be directly compared with those SWDs retrieved from GPS observations. To evaluate the performance enhancement of using multi-source water vapor data in tomography, we carry out a tomography using GPS water vapor data. For brevity, this





tomography is named as Tomo-I and the tomography using multi-source water vapor data is referred to as Tomo-II in this paper. For both Tomo-I and Tomo-II, the five vertical a priori information schemes are implemented and the corresponding results of tomographic wet refractivity field are evaluated.

Table 1 shows the self-consistency results obtained from different vertical a priori information schemes. The statistics are calculated from the differences between GPS inferred SWD/ZWD and tomography derived SWD/ZWD over the HKLT station (the evaluation GPS station). It can be seen that vertical constraint scheme V1 achieves the best performance in both Tomo-I and Tomo-II with RMS (root mean squares) errors of 11.30 mm and 10.85 mm for the slant wet delay (SWD), respectively. In addition, vertical constraint V3 performs better than V2, likewise for V5 and V4. This can be easily explained as: vertical constraint schemes V3 and V5 consider the variations of water vapor in different months but the vertical a priori information is invariable in both V2 and V4. Comparing the performance of Tomo-I with Tomo-II, we can observe that tomographic results from Tomo-II have higher accuracy than those from Tomo-I, except for the scheme V5. The Tomo-II with scheme V1 achieves the highest RMS accuracy of 6.46 mm in ZWD.

**Table 1: Statistics of the differences between GPS inferred SWD/ZWD and tomography derived SWD/ZWD over the HKLT station (unit: mm)**

| Vertical Constraint | Tomo-I | | | | Tomo-II | | | |
|---|---|---|---|---|---|---|---|---|
| | SWD | | ZWD | | SWD | | ZWD | |
| | Bias | RMS | Bias | RMS | Bias | RMS | Bias | RMS |
| V1 | -1.58 | 11.30 | 0.17 | 7.07 | -0.57 | 10.85 | -0.71 | 6.46 |
| V2 | 3.80 | 12.85 | -0.85 | 7.76 | 4.16 | 12.29 | -1.58 | 7.54 |
| V3 | 3.44 | 12.61 | -0.74 | 7.38 | 3.00 | 11.47 | -1.15 | 7.18 |
| V4 | 3.60 | 12.05 | -0.87 | 7.36 | 4.05 | 11.97 | -1.63 | 7.31 |
| V5 | 3.19 | 11.59 | -0.85 | 7.17 | 3.88 | 11.75 | -1.55 | 7.21 |

The tomographic results are also assessed using radiosonde vertical profile data. Statistical results of the differences of wet refractivity between radiosonde and tomography are presented in Table 2. The comparison results further demonstrate that V1 is the best vertical constraint scheme. As seen in Table 2, vertical constraint scheme V1 achieves an accuracy of 7.26 mm/km in Tomo-I and an even higher accuracy of RMS error of 7.13 mm/km in Tomo-II. For the other four schemes, their RMS errors range from 9.42 mm/km to 11.44 mm/km, clearly greater than the scheme V1. The tomographic results solved from schemes V3 and V5 are better than schemes V2 and V4, respectively. This is also consistent with the evaluation shown in Table 1 using GPS data. In Table 2, it is worth mentioning that for all the 5 schemes, tomographic results from Tomo-II are all consistently better than those of Tomo-I. Considering the results in both Table 1 and Table 2, we can conclude that scheme V1 is the best vertical constraint scheme. This reveals that averaging radiosonde profiles over a 3-day period as water vapor vertical a priori information is better than averaging over a longer period. That is to say it is better to employ recently observed radiosonde profiles as vertical a priori information in the tomography. In addition, it is demonstrated that



the assimilation of multi-source data into the water vapor tomography (Tomo-II) can improve the tomographic reconstruction accuracy over the tomography using GPS water vapor data only (Tomo-I).

**Table 2: Statistics of the differences of wet refractivity between radiosonde and tomography (unit: mm/km)**

| Vertical Constraint | Tomo-I | | Tomo-II | |
|---|---|---|---|---|
| | Bias | RMS | Bias | RMS |
| V1 | 0.71 | 7.26 | 0.85 | 7.13 |
| V2 | 1.00 | 11.29 | 1.31 | 10.01 |
| V3 | 1.22 | 9.85 | 1.37 | 9.64 |
| V4 | 0.96 | 11.44 | 1.36 | 9.73 |
| V5 | 1.39 | 10.53 | 1.36 | 9.42 |

The comparison analysis presented above shows the overall accuracy along a slant or zenith path but does not show the accuracy of tomographic results at different layers. To study the tomographic accuracy at different altitudes, the RMS errors and the relative RMS errors of the differences between radiosonde and tomography at different layers are calculated. Relative RMS error is defined as the radiosonde measured wet refractivity divided by RMS error. Figure 4 shows the change of RMS error and relative RMS error with altitude for 10 different scenarios defined in Table 1. Generally, the RMS error

decreases with the increase of altitude due to the water vapor content decreases with the altitude. For the best scenario Tomo-II_V1, its RMS error is 11.44 mm/km at the lowest layer (0 to 0.4 km) and decreases to 3.30 mm/km at the uppermost layer (7.5 to 8.5 km). In terms of the relative RMS error, its value increases from 9% at the lowest layer to 67% at the uppermost layer for Tomo-II_V1, revealing the deficiency of tomography in retrieving the water vapor of high altitude layers. Generally speaking, tomographic wet refractivity fields solved by Tomo-II (curve with solid square) are better than those

derived by Tomo-I (dashed line with hollow triangle) at most of the layers. For the scheme V1, Tomo-II shows slightly better performance than Tomo-I at all layers. Referring to the other four schemes, it can be seen from Figure 4 that tomographic results solved from Tomo-II are significantly better than those from Tomo-I especially in the lower layers. This clearly demonstrates the positive contribution of multi-source water vapor data to the water vapor tomography. As indicated before, the four schemes V2 to V5 are probably too coarse to characterize the vertical variation of the water vapor.

Especially a flat region like Hong Kong, the accuracy of tomography is highly dependent on the accuracy of vertical a priori information.





**Figure 4: RMS errors (a) and relative RMS errors (b) of the differences between wet refractivity derived from radiosonde and tomography on different altitude layers during May to October 2013**



## 4.2 Capability of the tomography under conditions of heavy precipitation

The overall performance of the water vapor tomography using multi-source data is evaluated in the last section. It should be noted that one of very important goals of water vapor tomography is to provide accurate 3D water vapor data and information to support weather forecasting under heavy precipitation conditions. The heavy precipitation is defined as accumulated rainfall exceeding 30 mm within one hour. During the study period May to October 2013, a total of 15 days occurred heavy precipitation events. This section will focus on tomographic accuracy assessment under heavy precipitation conditions. The last section demonstrates that the scheme Tomo-II_V1 can achieve the highest tomographic accuracy, thus only this scheme is used in the performance assessment in this section.

As seen in Table 3, the RMS error of the differences between tomographic SWD and GPS inferred SWD is 10.98 mm under conditions of heavy precipitation. For the comparison between tomography and radiosonde, an RMS error of 7.36 mm/km is yielded. It can be noted that their RMS errors are slightly larger than the overall RMS errors shown in the previous sections. This is due to the fact that water vapor is much more dynamic and abundant under heavy precipitation conditions. Nevertheless, the tomography still achieves a good accuracy during heavy precipitation conditions. This demonstrates that the robustness of this water vapor tomographic modeling software system and that only slight degradation in water vapor tomographic accuracy can be observed under heavy precipitation conditions.

**Table 3: Comparison of tomography with GPS and radiosonde under conditions of heavy precipitation during May to October 2013. Tomography is carried out using multi-source data with vertical constraint scheme V1.**

| Tomography *vs* GPS (mm) | | Tomography *vs* Radiosonde (mm/km) | |
|---|---|---|---|
| Bias | RMS | Bias | RMS |
| 2.25 | 10.98 | 1.17 | 7.36 |

During 21~22 May 2013, a torrential storm occurred in Hong Kong with daily rainfall of 190 mm, which was the maximum daily rainfall over the past five years. On that day, HKO issued the highest level of warning signal – black rainstorm (black rainstorm signal means heavy rain exceeding 70 mm in an hour). The rainstorm lasted about 9 hours from 17:00 UT 21 May to 02:00 UT 22 May 2013. Water vapor tomographic technique provides us a powerful tool to investigate the spatio-temporal characteristics of the water vapor variability for this severe convective weather. By using the tomographic wet refractivity and pressure and temperature data provided by NWP model, the partial pressure of water vapor can be solved from Eq. (3) and the relative humidity field could be further determined. Figure 5(a) presents the evolution of tomographic relative humidity profiles at the HKO weather station (22.30 °N, 114.17 °E) during the period from 04:00 UT 21 May to 10:00 UT 22 May 2013. The evolution of ZWD measurements is shown in Figure 5(b) along the same time series. Taking the total ZWD as a focus first, it can be observed that total ZWD values continuously increased from ~340 mm at 07:00 UT 21 May to ~400 mm at 17:30 UT 21 May when the precipitation started. After then the total ZWD shows a small decrease followed by a quick increase. When the total ZWD peaked at 19:30 UT 21 May, the torrential rain came. With the rain



downpouring, the total ZWD decreased quickly. In the five following hours from 21:00 UT 21 May, the total ZWD fluctuated while the heavy rain weakened to drizzles. It can also be seen that the total ZWD shows a quick decrease after the end of this precipitation event. Examining the tomographic relative humidity profiles in Figure 5(a) can help us to better understand the spatio-temporal variation of the water vapor during the rainstorm. We can find that the change of ZWD is

5    mainly attributed to the variation of water vapor at lower layers. Especially the water vapor below 3 km showed evident fluctuations. During the rainstorm, relative humidities for layers below 2 km and 3~5 km were very high approaching 100%, indicating that there was abundant water vapor to fuel the heavy rain. In addition, the ZWD variations at five layers are also given in Figure 5(c). The ZWD below 1 km reached the maximum at 18:00, 21 May 2013 when the rain just began. Then the ZWD below 1 km decreased quickly during the heavy precipitation. ZWDs between 1~2 km kept in a steady status and did

10   not show much fluctuations. We can observe that the increase of the total ZWD during 18:00~20:00, 21 May 2013 is mainly attributed to the layers between 2~5 km. At the same time, water vapor above 5 km showed a slow decrease followed by a sudden increase. In the subsequent five hours (21:00 UT 21 May to 02:00 22 May) after the heavy precipitation, small rains continued. Water vapor in each layer still showed much fluctuations during this period. This indicated that the atmosphere was in an unstable condition and precipitations continued to occur. Once the precipitation ended, it could be found that water

15   vapor in different layers gradually restored towards a steady state.



**Figure 5: Evolution of tomographic relative humidity profiles (a), total ZWD (b) and ZWD at various layers (c) every 30 minutes from 04:00 UT 21 May to 10:00 UT 22 May 2013 over the HKO weather station. The blue lines in subfigure (a) show the gauged rainfall within each 30 minutes and their values correspond to the right vertical axis. Subfigure (c) presents the ZWD below the height of 1000 m (blue curve with circle), ZWD between 1000 m and 2000 m (pink curve with square), ZWD between 2000 m and 3000 m (yellow curve with triangle), ZWD between 3000 m and 5000 m (green curve with rhomb), and ZWD above the height of 5000 m (red curve with inverse triangle).**

A more detailed illustration of the evolution of tomographic relative humidity profiles can be found in Figure 6. Subgraphs tagged with 'a' and 'b' refer to the relative humidity sections along the longitude of 114.17 °E (south-north section) and latitude of 22.30 °N (west-east section), respectively. In Figure 6(1a), (1b), (2a), (2b), (3a), (3b), (8a) and (8b) the relative



humidity profiles show relatively steady conditions. In Figure 6(4a), (4b), (5a), (5b), 6(a), 6(b), (7a) and (7b) we can observe there are some disturbances of relative humidity implying the instability of the atmosphere. Especially in Figure 6(5a) and (5b), large disturbances exist (relative humidities in most layers approach 100% and in some upper layers close to 0) and we know that at this time torrential rain was pouring. It should be noted that one of the prerequisites of forming a convective

5    storm system is the existence of enough moisture in the lower to mid troposphere. The tomographic water vapor distribution shown in Figure 5 and Figure 6 indicates that there was abundant water vapor in the lower troposphere. This water vapor tomographic example during a typical rainstorm illustrates that the tomographic technique can well reveal the spatial structure and temporal variation of the atmospheric water vapor under rainstorm conditions.

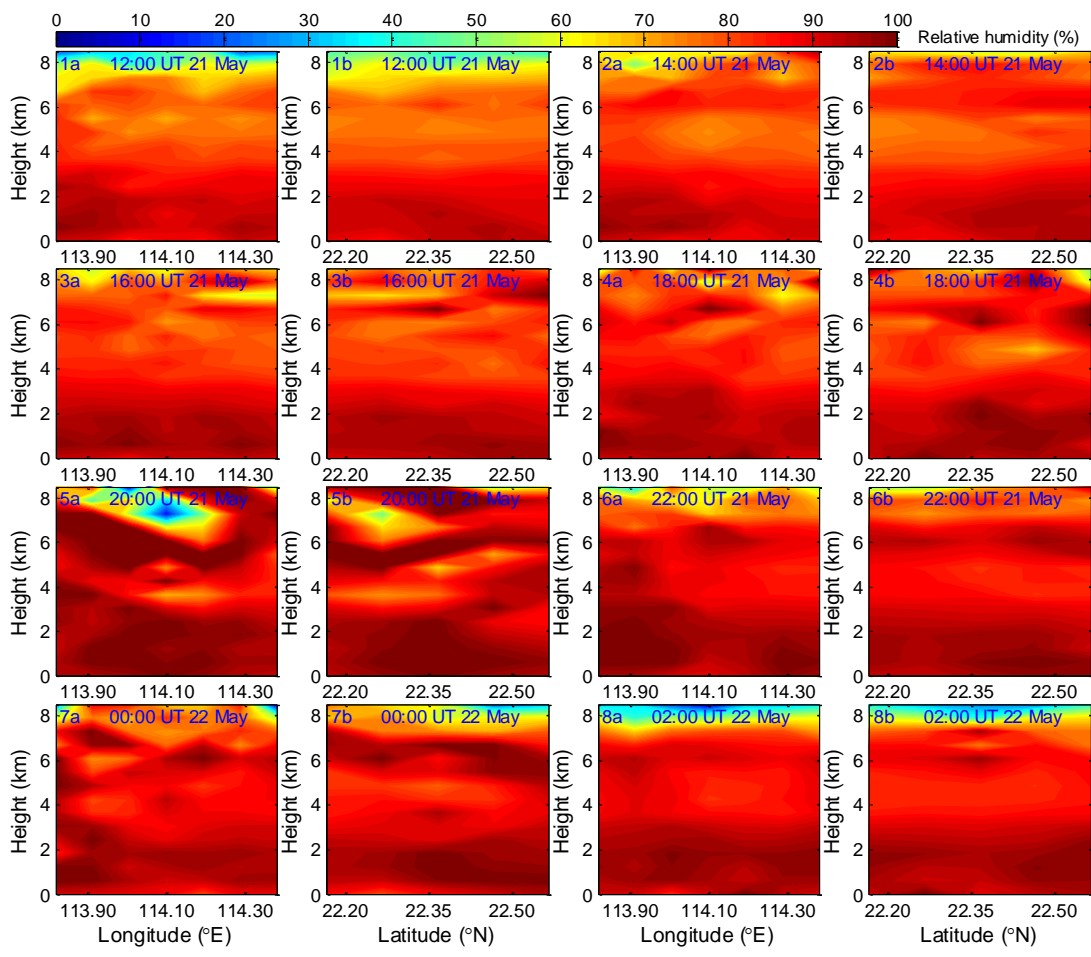

**Figure 6: Evolution of tomographic relative humidity profiles every 2 hours from 12:00 UT 21 May to 02:00 UT 22 May 2013. Subgraphs tagged with 'a' refer to the relative humidity sections along the longitude of 114.17 °E. Subgraphs tagged with 'b' refer to the relative humidity sections along the latitude of 22.30 °N. The longitude 114.17 °E and latitude 22.30 °N specify the location of the HKO weather station.**





## 5 Discussion and conclusion

As a crucially important atmospheric parameter, accurate water vapor data in the spatial and temporal domains can play a significant role in the study of many atmospheric processes. Water vapor tomography has been proven a potent technique that is capable to retrieve the spatio-temporal distribution of the atmospheric water vapor. Traditionally, water vapor

tomography is often performed by using water vapor measurements derived from GPS/GNSS observations. The integration of GPS-derived and other sensors' water vapor data in principal can augment the tomographic modeling system and improve the water vapor modeling accuracy.

Based on this idea, this paper develops a multi-source water vapor tomographic modeling system in Hong Kong by using water vapor data collected from GPS, radiosonde, WVR, NWP, AERONET sun photometer and meteorological instruments.

The radiosonde data are not directly employed. Instead, they are used to provide vertical a priori information for the tomography. Five different vertical constraint schemes are examined in this study. To show the performance, tomography results using multi-source data (Tomo-II) are compared against those using GPS water vapor data only (Tomo-I), using six months' data collected from May to October 2013.

Tomographic results are assessed with water vapor data derived from both GPS and radiosonde. It shows that the scheme V1

of using vertical a priori information derived from three days of radiosonde observations prior to the tomographic epoch achieves the best performance in both Tomo-I and Tomo-II. With the use of the best vertical a priori information (scheme V1), the Tomo-II strategy has the following performance: (1) SWD data achieve an accuracy of 10.85 mm when assessed by GPS inferred SWD measurements; (2) the whole wet refractivity profiles yield an RMS error of 7.13 mm/km when assessed by radiosonde observed wet refractivity ones; (3) in terms of accuracy along vertical layer, RMS errors generally decrease

with altitude from 11.44 mm/km at the lowest layer (0 to 0.4 km) to 3.30 mm/km at the uppermost layer (7.5 to 8.5 km). The corresponding relative RMS errors increase from 9% to 67%, revealing the deficiency of tomography in retrieving the water vapor of high altitude layers.

Water vapor tomography using the best tomographic scheme is further evaluated under heavy precipitation conditions in Hong Kong. Analysis results show that tomography performance during the rainstorm period is only slightly degraded

compared to that in the whole evaluation period May to October 2013. The tomography results during the 21~22 May 2013 rainstorm show that atmospheric water vapor content increases prior to the occurrence of the rainstorm and decreases during the precipitation. This experiment indicates that the water vapor tomography result may make a contribution to the forecasting of severe weather conditions.

### Acknowledgments

This work is supported by National Natural Science Foundation of China (No. 41274039), the Hong Kong Research Grants Council (RGC) Early Career Scheme Project (PolyU 5325/12E, F-PP0F), General Research Fund Project (PolyU 5203/13E, B-Q37X) and the Hong Kong Polytechnic University project (PolyU 152168/15E, G-YBM3). The authors would like to





appreciate the help of Mr. Wai Kin Wong, Mr. Wang Chun Woo, Mr. Sai Tick Chan, and Mr. P. W. Chan from the Hong Kong Observatory, the Government of Hong Kong Special Administrative Region (HKSAR) for providing the WVR and NWP data. The Lands Department of the Government of HKSAR is thanked for providing the SatRef (Hong Kong Satellite Positioning Reference Station Network) GPS data. Many thanks also go to Professor Janet Elizabeth Nichol from the

Department of Land Surveying & Geo-Informatics, the Hong Kong Polytechnic University for providing AERONET sun photometer data. The Department of Atmospheric Science of the University of Wyoming is acknowledged for providing the Hong Kong radiosonde data.

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
