# Peer review of "Assessing the Performance of Troposphere Tomographic Modeling Using Multi-Source Water Vapor Data During Hong Kong's Rainy Season from May to October 2013"

_Atmospheric Measurement Techniques, 2016_

## Referee Comment (RC1) · Anonymous Referee #1 · 25 Aug 2016

General comments: This paper presents the proposal of troposphere modelling using tomography technique and multi-source water vapor data. The integration method of different observation is the novel approach and gives the possibility to improve the stability of equation system inversion. The results are significant and verified independently.

The answers to the main question for the reviewer:

1. Does the paper address relevant scientific questions within the scope of AMT? Yes 2. Does the paper present novel concepts, ideas, tools, or data? Yes, the propose of data

integration in one solution. 3. Are substantial conclusions reached? Yes 4. Are the scientific methods and assumptions valid and clearly outlined? Yes/No details comments below 5. Are the results sufficient to support the interpretations and conclusions? Yes 6. Is the description of experiments and calculations sufficiently complete and precise to allow their reproduction by fellow scientists (traceability of results)? Yes 7. Do the authors give proper credit to related work and clearly indicate their own new/original contribution? Yes 8. Does the title clearly reflect the contents of the paper? Yes 9. Does the abstract provide a concise and complete summary? Yes 10. Is the overall presentation well structured and clear? Yes 11. Is the language fluent and precise? Yes 12. Are mathematical formulae, symbols, abbreviations, and units correctly defined and used? Yes 13. Should any parts of the paper (text, formulae, figures, tables) be clarified, reduced, combined, or eliminated? No 14. Are the number and quality of references appropriate? Yes 15. Is the amount and quality of supplementary material appropriate? Yes

Questions and comments:

p.9 l. 5 What observations were processed GPS or GNSS? It is not clearly explained.

p.9 l.20 The procedure for determining the weights for observations require stronger justification. The process of balancing equations observation weights should be the result of the analysis of the accuracy of observations. In paper for three types of observations unit weights are used and why they are equivalent?

---

## Author Comment (AC1) · 9 Sep 2016

Please find the reply and revised manuscript in the attached zip file. Thank you very much.

Please also note the supplement to this comment:
http://www.atmos-meas-tech-discuss.net/amt-2016-158/amt-2016-158-AC1-supplement.zip

---

## Referee Comment (RC2) · Anonymous Referee #2 · 23 Sep 2016

Review of the paper: "Assessing the Performance of Troposphere Tomographic Modeling Using Multi-Source Water Vapor Data during Hong Kong's Rainy Season from May to October 2013"

Authors: Biyan Chen, Zhizhao Liu

The paper deals with an another water vapour observation system (tomography) based on GNSS data. The original aspect of the paper is the assimilation of different kind of water vapour measurements (radio-sounding, photometers, radiometers). Such an assimilation of water vapour measurement is a classical approach (even if it should

be taken into account the lack of vertical resolution of the GNSS data). The main interesting impact of the paper that should be reinforced is to clearly state the added value of GNSS data.

The paper is well written and scientifically sounded. The structure of the paper could be enhanced (some repetition in p.8 for example; the description of the inversion is quite long and do not detail if a temporal link is included between the inversions; it is not clear what kind of measurements are assimilated in the NWP models used).

To enhance the impact of the paper, 2 points should be added. (1) To clearly establish the added value of the GNSS data, some others comparison could be done as some comparison between the a priori model (maybe only the best one: V1) and the radiosounding or GNSS SWD as in Champollion et al., 2002. (2) From the water vapour observation system (tomography) based on GNSS data, it is not clear to see the impact of data versus NWP models. Could the authors test the tomography without the NWP models assimilation? It is also not clear if the authors also test the assimilation of the surface humidity data.

The case study during severe precipitations event is a nice illustration of the potential of the GNSS tomography. Is the difference between the NWP model and tomography increasing during such a high precipitation event? Even if this is out of the scope of the paper, the study should extended with additional data such as wind to focus on atmospheric processes.

Reference: Champollion, C., Flamant, C., Bock, O., Masson, F., Turner, D. D., & Weckwerth, T. (2009). Mesoscale GPS tomography applied to the 12 June 2002 convective initiation event of IHOP_2002. Quarterly Journal of the Royal Meteorological Society, 135(640), 645-662.
* * *